# Sparse Shortcuts: Facilitating Efficient Fusion in Multimodal Large Language Models

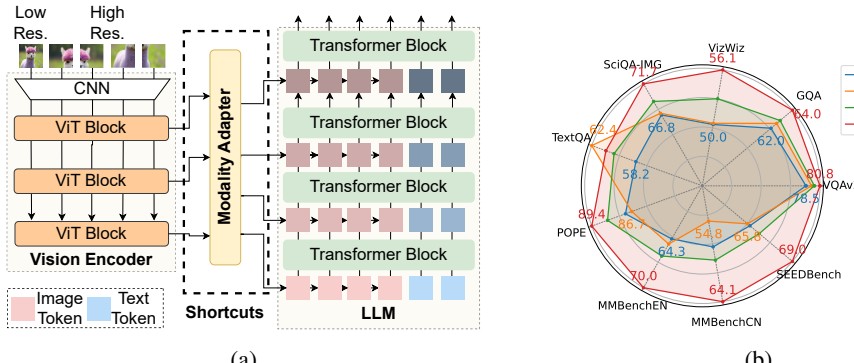

Figure 1: (a) The overview of the SparseCut mechanism. It supports multi-level cross-modal fusion across different model layers, as well as high- and low-resolution image fusion through shortcut connections. (b) Performance of SparseCut on various multimodal benchmarks.

## Abstract

With the remarkable success of large language models (LLMs) in natural language understanding and generation, multimodal large language models (MLLMs) have rapidly advanced in their ability to process data across multiple modalities. While most existing efforts focus on scaling up language models or constructing higher-quality training data, limited attention has been paid to effectively integrating cross-modal knowledge into the language space. In vision-language models, for instance, aligning modalities using only high-level visual features often discards the rich semantic information present in mid- and low-level features, limiting the model's ability of cross-modality understanding. To address this issue, we propose SparseCut, a general cross-modal fusion architecture for MLLMs, introducing sparse shortcut connections between the cross-modal encoder and the LLM. These shortcut connections enable the efficient and hierarchical integration of visual features at multiple levels, facilitating richer semantic fusion without increasing computational overhead. We further introduce an efficient multi-grained feature fusion module, which performs the fusion of visual features before routing them through the shortcuts. This preserves the original language context and does not increase the overall input length, thereby avoiding an increase in computational complexity for the LLM. We systematically evaluate the performance of various shortcut patterns and demontrate that SparseCut can enhance the performance of MLLMs across various multimodal benchmarks with high training stability. It is also compatible with different base LLMs. [1]

## 1 Introduction

In recent years, large language models (LLMs), such as ChatGPT (Achiam et al., 2023a), have achieved remarkable progress in natural language understanding and generation, spurring rapid developments in multimodal large language models (MLLMs) (Kuang et al., 2025). By integrating

---

[1]The code is available in the supplementary material for blind review.

information from multiple modalities—such as vision and audio—MLLMs exhibit strong cross-modal understanding and reasoning capabilities, delivering impressive performance on tasks including image-text matching (Chen et al., 2025), visual question answering (Kim et al., 2025), and instruction following (Dong et al., 2025). These advances have attracted significant attention from both academia and industry.

Focusing on visual-language models, the prevailing architecture of MLLMs typically consists of three core components: a *vision encoder*, a *modality adapter*, and an *LLM*. The vision encoder extracts visual features; the modality adapter aligns these features with the language space; and the LLM fuses the aligned visual and textual representations for multimodal understanding and generation. While prior studies have comprehensively covered architectural innovations (Zhao et al., 2023), scaling strategies (Biderman et al., 2023), and instruction tuning (Huang et al., 2023), relatively few works have explored how to efficiently transfer visual source information into the language space.

Current MLLMs face limitations in leveraging visual semantics effectively. First, most models rely solely on the final-layer output of a pretrained vision encoder—such as CLIP-ViT (Radford et al., 2021) or SigLIP (Zhai et al., 2023)—for cross-modal alignment (Yin et al., 2024), neglecting the rich multi-level semantics captured at intermediate layers. Transformer-based vision encoders, such as ViT (Dosovitskiy et al., 2020), encode hierarchical features: shallow layers capture local textures, middle layers focus on structures and relations, and deeper layers encode global semantics. By discarding low- and mid-level representations, existing models forgo valuable visual cues essential for fine-grained multimodal reasoning. Second, although incorporating multi-grained visual features, i.e., features at multiple spatial resolutions, has shown promise (Fei et al., 2024; He et al., 2024), it substantially increases computational cost. Appending multi-resolution visual tokens inflates the input sequence length to the LLM, causing a quadratic increase in attention computation.

To address these limitations, we propose **SparseCut**, a general cross-modal fusion architecture of MLLMs supporting efficient multi-level and multi-grained feature utilization (Fig. 1a). SparseCut incorporates multiple shortcut connections between different layers of the vision encoder and LLM, promoting the structural integration of multi-level visual features and textual features. These shortcut connections enable the LLM to access rich visual semantic information conveyed by multi-level visual hidden states at various depths, enhancing the model's capability to cross-modal understanding and generation without significantly increasing computational overhead. We discuss various factors defining a shortcut pattern, including the order, distribution, and density of connections. The current MLLM architecture is a special case of SparseCut. To further exploit multi-grained visual features through the shortcut connections, we propose to fuse low- and high-resolution visual features in the transmission through each shortcut connection. Compared to existing multi-grained approaches that append multi-resolution visual feature tokens to textual tokens, our proposed approach does not increase the overall input context length, avoiding quadratic growth in computational overheads in the LLM. Experiments on multiple benchmarks demonstrate that ShortCut significantly improves performance in various multimodal tasks (as depicted in Fig. 1b), exhibiting generality across different base LLMs and scalability in LLMs of different sizes. The main contributions of this paper are as follows:

- We introduce SparseCut, a general MLLM cross-modal fusion architecture with sparse shortcut connections. SparseCut integrates multi-level hidden states of different modalities through a structural connection pattern.
- SparseCut efficiently fuses multi-grained visual features for the input of LLM through the shortcuts, without increasing the LLM context length and computational complexity.
- We systematically evaluate various shortcut patterns. SparseCut improves multimodal task performance with high training stability and is compatible with different base LLMs.

## 2 RELATED WORK

### 2.1 LARGE LANGUAGE MODELS

The rapid advancement of autoregressive large language models (LLMs) (Brown et al., 2020; Chowdhery et al., 2023; Achiam et al., 2023b) has drawn significant attention from both academia and industry. Numerous LLMs (Touvron et al., 2023a; Zhang et al., 2022; Touvron et al., 2023b) have since emerged, with open-source initiatives like LLaMA (Touvron et al., 2023b) playing a pivotal role in accelerating research and fostering broader community participation.

Instruction tuning (Ouyang et al., 2022) has enabled these models to align better with human intent, demonstrating strong generalization across a wide range of natural language understanding and generation tasks. Recent development trends follow a dual trajectory: compact models such as Alpaca (Taori et al., 2023), Phi (Abdin et al., 2024), and Mistral (Jiang et al., 2023) target resource-efficient deployment, while large-scale models like GPT-4 (Achiam et al., 2023b) and PaLM (Chowdhery et al., 2023) continue to scale in size to push the limits of reasoning and generative capabilities.

## 2.2 MULTIMODAL LARGE LANGUAGE MODELS

Following the success of the Transformer architecture in the language domain, Google pioneered its application in computer vision by proposing the Vision Transformer (ViT) (Dosovitskiy et al., 2020), which achieved remarkable results across various visual tasks and validated the potential of Transformer-based models for visual representation learning. Building upon this, OpenAI introduced the CLIP model (Radford et al., 2021), which aligned images and text within a unified contrastive learning space. Trained on large-scale image-text pairs, CLIP demonstrated strong zero-shot and few-shot performance on numerous downstream tasks, establishing a significant milestone in multimodal pretraining.

Subsequently, researchers began exploring how to equip large language models (LLMs) with visual understanding capabilities. Models like BLIP-2 (Li et al., 2023b) and InstructBLIP (Dai et al., 2023) further advanced this paradigm by more deeply integrating the perception mechanism with LLMs, significantly improving performance in tasks such as image captioning and visual question answering. Representative works such as Flamingo (Alayrac et al., 2022) and BLIP (Li et al., 2022) adopted the visual perceiver architecture, employing cross-attention mechanisms to resample visual features and generate visual tokens that are input into the LLM. However, these visual perceptron-based approaches often rely on large-scale datasets, resulting in slow training convergence and limited generalization in diverse downstream tasks.

LLaVA (Liu et al., 2024a) proposed a more efficient architecture by introducing an MLP adapter to directly bridge the pretrained vision encoder and the LLM. This architecture reduces training and inference costs while achieving competitive performance across multiple benchmarks, and has gradually evolved into a mainstream paradigm for multimodal large models. Around this design, subsequent works have continued to refine image resolution handling and token compression strategies to further improve computational efficiency and representational effectiveness. This stage's typical architecture can thus be summarized as: *frozen vision encoder + adapter + LLM*. This approach relies solely on the final-layer output of the vision encoder, thus neglecting the wealth of fine-grained visual details contained in earlier layers. To address this limitation, we aggregate hidden features from different layers of the vision encoder and progressively fuse them from deeper to shallower layers, aiming to recover the lost semantic richness and enhance the model's visual understanding capability.

More recent studies have introduced innovations along two primary axes: token compression and multi-granularity fusion. For token compression, LLaMA-VID (Li et al., 2023c) proposed a dual-token representation strategy that generates content and context tokens to significantly compress visual input, enabling efficient processing of extended video sequences while maintaining strong performance across text-video tasks. DeepStack-VL (Meng et al., 2024) explores multi-granularity fusion by splitting high-resolution images into patches, encoding each sub-image independently, and injecting both high- and low-resolution visual information into the bottom layers of the language model through residual connections. However, this layered injection strategy still does not leverage intermediate representations from the vision encoder. Qwen3-VL (Qwen Team, 2025), released recently, integrates multi-level features from ViT by injecting representations from intermediate ViT layers into several bottom layers of the language model via a DeepStack-style fusion mechanism. This enhancement aligns with our core idea of capturing multi-level visual details for improved vision–language alignment. Our method designs a general architecture for multi-level multi-grained cross-modal fusion, systematically emphasizing key characteristics, including connection order, connection-end distribution, and density.

# 3 METHOD

## 3.1 OVERVIEW

Fig. 1a illustrates the overview of the ShortCut mechanism within the mainstream LLaVA (Liu et al., 2024a) framework. It includes a vision encoder, a modality adapter, an LLM, and multiple shortcuts connecting different layers between the vision encoder and the LLM.

**Vision Encoder:** It is typically a vision Transformer-based (ViT-based) encoder (Dosovitskiy et al., 2020), such as the pretrained CLIP (Radford et al., 2021), with multiple layers to extract visual features of different level of semantics. The input image $X$, including its low- and high-resolution formats, is first divided into $N$ non-overlapping patches. Each patch is then processed via a convolutional layer into a sequence of visual tokens, forming the initial sequence of visual hidden states $X_0 \in \mathbb{R}^{N \times M_v \times D_v}$, where $M_v$ is the visual token sequence length and $D_v$ is the hidden size of ViT. This sequence is fed into ViT, consisting of $L_v$ layers. The $i$-th ViT layer, $V_i(\cdot) : \mathbb{R}^{N \times M_v \times D_v} \rightarrow \mathbb{R}^{N \times M_v \times D_v}$, encodes a sequence of hidden states into a new sequence of the same shape:

$$X_i = V_i(X_{i-1}). \tag{1}$$

**Modality Adapter:** It fuses multi-grained visual features and aligns visual features with the textual semantic space of the LLM. Specifically, it first applies cross-attention between the hidden states of high- and low-resolution images for each layer that has a shortcut connection to the LLM. For the visual hidden states $X_i^{high}$ and $X_i^{low}$ output by the $i$-th ViT layer, the fused visual tokens after cross-attention is:

$$Y_i = attn(X_i^{low}, X_i^{high}), Z_i = MLP(Y_i) \tag{2}$$

where low-resolution tokens $X_i^{low}$ serve as the *query* and high-resolution tokens $X_i^{high}$ serve as the *key* and *value* of cross-attention. Note that this process reduces the visual hidden states in the token dimension. The resulting visual tokens $Y_i \in \mathbb{R}^{M_v \times D_v}$ are then fed into a feedforward network to further align with the textual semantic space. This yields $Z_i \in \mathbb{R}^{M_v \times D_t}$, where $D_t$ is the hidden size of the LLM.

**LLM:** The LLM initially uses a tokenizer and text embedding module to transform text input data into a sequence of textual tokens $T_0 \in \mathbb{R}^{M_t \times D_t}$, where $M_t$ is the textual sequence length. The LLM treats visual tokens as textual tokens and concatenates them with other language textual tokens as the Transformer input. Let $Z_j'$ and $Z_j'' \in \mathbb{R}^{M_v \times D_t}$ be the sequences of visual tokens corresponding to the input and output of the $j$-th LLM Transformer layer, respectively. We have:

$$[Z_j'', T_j] = Transformer_j([Z_j', T_j]). \tag{3}$$

The shortcut connections potentially affect the input visual tokens of a Transformer layer in the LLM. Let $S = (i, j)$ be the set of shortcut connections between the $i$-th ViT layer and the $j$-th LLM layer. For an LLM layer without a shortcut connection, the input visual tokens are the corresponding output of the previous layer, the same as conventional MLLMs. For an LLM layer with a shortcut connection, the input visual tokens are the sum of the corresponding output of the previous layer and the fused visual tokens from the connecting ViT layer. Therefore, we have

$$Z_j' = \begin{cases} Z_{j-1}'' & \text{if } (i, j) \notin S, \forall i \\ Z_{j-1}'' + Z_i & \text{if } (i, j) \in S, \exists i \end{cases} \tag{4}$$

The above procedure defines a general MLLM framework supporting multi-level and multi-grained fusion with shortcuts. Specifically, the special case of $S = \{(L_v, 1)\}$ is the conventional MLLM architecture. The only connection from the last layer of the vision encoder to the first layer of the LLM enables projecting the final vision output as the LLM input. Next, we draw the focus onto the construction of the shortcut connection set $S$, which indicates a shortcut pattern.

## 3.2 SHORTCUT CONNECTIONS

A shortcut connection pattern can be identified by three factors: connection order, connection-end distribution, and density. Combinations of different settings of these factors result in different shortcut patters, as shown in Fig. 2.

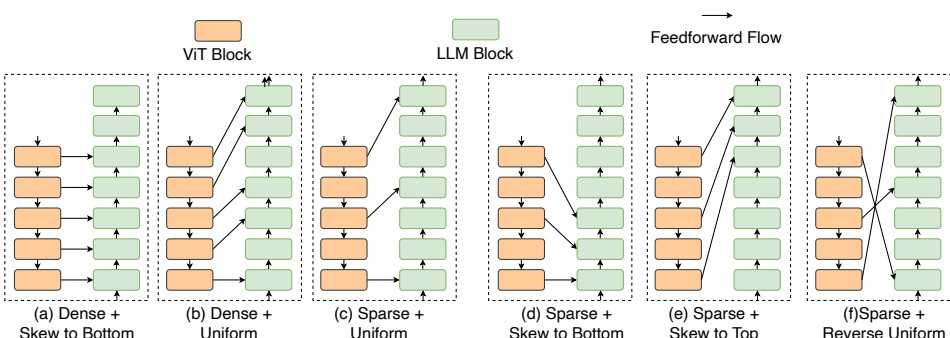

Figure 2: Various shortcut connection patterns

**Connection order:** It refers to the relative depth alignment between the connected ViT and LLM layers. Inspired by the design principle of U-Net (Ronneberger et al., 2015), SparseCut can employ a U-shape connection order, where shallow (low-level) visual layers are connected to deep (high-level) language layers, and vice versa. Formally, for any two shortcut connections $(i_1, j_1)$, $(i_2, j_2) \in S$, the U-shape constraint is defined as: $i_i > i_2$ if and only if $j_1 < j_2$. An aligned-depth alternative is reversing the U-shape to connect shallow visual layers with shallow language layers and deep visual layers with deep language layers: $i_i > i_2$ if and only if $j_1 > j_2$. Figs. 2(a)-(e) illustrate various such U-shape shortcut patterns and Fig 2(f) illustrates the aligned-depth order.

This structured mapping of U-shape connections, forming a cross-level integration between visual and textual representations, may have potential advantages over the aligned-depth alternative. The U-shape pattern encourages complementary information to flow bidirectionally across modalities. For example, low-level visual details have highways to inform high-level language reasoning. While multiple aligned skip connections of the aligned-depth order may introduce potential redundancy and offer diminishing returns.

**Connection-end distribution:** It describes how the shortcut connections are spread across the LLM layers. Figs. 2(c)-(e) illustrate shortcut connections with uniform distribution, skewed to the LLM bottom, and skewed to the LLM top, respectively. The right end of shortcuts in Fig. 2(c) spans evenly across the entire range of the LLM, influencing all depths of language processing. The shortcuts in Figs. 2(d) and 2(e) concentrate at the bottom and top layers of the LLM, respectively, favoring the cross-modal integration with lower-level or higher-level textual features.

The intuition is that the uniform distribution allows the language model to integrate hierarchical visual context in a balanced way, giving the LLM a higher opportunity to reconcile the semantic mismatch. The bottom-skewed distribution can maximize early-stage integration of visual information, thereby granting the language model greater flexibility in modeling foundational visual semantics. In the case of the top-skewed distribution, all visual information is injected into only the top few LLM layers, bypassing most LLM decoder layers and preventing the model from leveraging its full hierarchical reasoning capacity. Consequentyl, the LLM may not adequately process or refine the visual representations, potentially resulting in substantially weaker image understanding.

**Density:** It indicates the level of overall density of shortcut connections, i.e., the number of connections in $S$ over the total number of ViT layers (which is typically lower than that of LLM layers). Dense patterns promote more extensive cross-modal integration, whereas sparse patterns focus information flow through a few critical paths, as illustrated in Figs. 2(b) and 2(c), respectively.

The sparse pattern has potential advantages over the dense one. Dense connections may introduce "information flooding". Although dense connections intuitively provide richer information fusion, in practice, they can lead to redundancy, interference, or even training instability, particularly in smaller models with limited parameter budgets. In contrast, sparse connections can offer a more efficient and generalizable mechanism for feature injection by reducing unnecessary redundancy while preserving critical structural signals across layers. This approach reflects a selective cross-modal feature mapping design: rather than aligning all visual features simultaneously, features should be injected selectively, spatially, and semantically, based on the specific needs of the language model at each decoding stage. The sparse pattern also requires lower computational overheads.

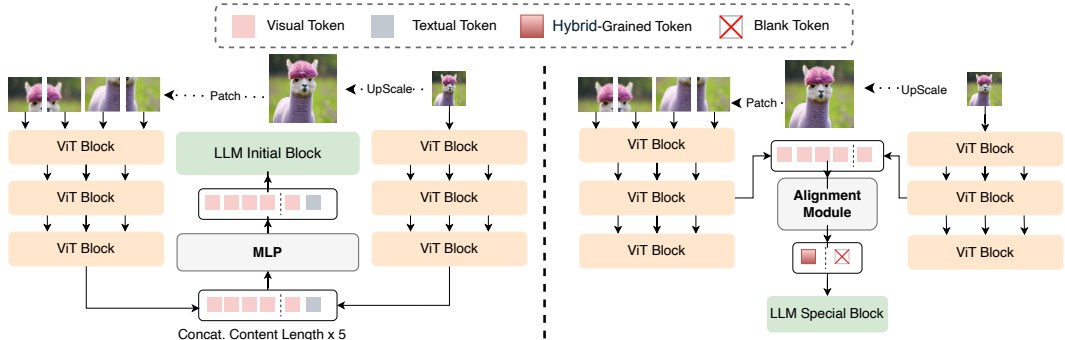

Figure 3: A comparison of different multi-grained visual fusion in an MLLM. (a) Traditional fusion: Conventional MLLMs handle high-resolution features by simply concatenating them with low-resolution ones along the token dimension, resulting in an image token sequence that is five times longer. (b) Shortcut fusion: Multi-resolution features fuse in each shortcut to the LLM, reducing the visual context length by four times. The blank token corresponds to textual tokens, meaning that no operation is performed on original textual tokens at this stage.

### 3.3 EFFICIENT MULTI-GRAINED FUSION

Adapting to inputs of varying granularity has become a critical challenge in the development of MLLM (Dehghani et al., 2023). Strict input constraints, such as fixed image resolutions, can restrict the expressive abilities of LLMs. In addition to leveraging multi-level semantic features from different layers of the ViT, we further enhance input-level granularity by introducing visual diversity directly into the model's inputs, enabling richer and more flexible visual representation.

SparseCut supports efficient multi-grained visual fusion through the cooperation of the vision encoder and modality adapter. As illustrated in Fig. 3, the original image is first upsampled via simple resizing, then divided into $N-1$ sub-images that match the Vision Transformer (ViT)'s input size requirements. These high-resolution sub-images, along with the original low-resolution image, forming $N$ (five in this paper) image patches and are independently encoded. Compared to using only the low-resolution input, integrating high-resolution input increases the number of visual tokens by a factor of four. Instead of fusing multi-grained features by using an MLP to reduce the hidden size of visual tokens to align with the LLM (Fig. 3(a)), SparseCut merges multi-grained visual tokens in the resolution dimension and generates hybrid-grained visual tokens to reduce visual context length (Fig. 3(b)). This is achieved by the cross-attention between low- and high-resolution visual tokens before an MLP, as introduced before.

The computational efficiency gain of the multi-grained fusion can be remarkable. The reason is that the modality adapter maintains the same visual context length as the approach with only low-resolution input. Let $M_v$ represent the visual context length for each image patch in ViT and $M_t$ be the textual context in the LLM. When the conventional multi-grained fusion solution generates a sequence of $N \times M_v$ visual tokens for the integration with textual tokens in the LLM, SparseCut generates only $M_v$ visual tokens. Given the quadratic complexity of the self-attention mechanism of the Transformer, this design reduces the computational complexity from a factor of $\mathcal{O}((N \times M_v + M_t)^2)$ to a factor of $\mathcal{O}((M_v + M_t)^2)$. For a concrete example, after integrating SparseCut in Llava-1.5 (Liu et al., 2024a), the FLOPs of vanilla LLaVA and SparseCut are almost identical (about 8.04T) when using only low-resolution image tokens. However, in the case when mixing with high-resolution image tokens, SparseCut consumes significantly lower FLOPs than vanilla LLaVA (9.6T v.s. 43.62T) due to the resolution-dimension visual token fusion mechanism of SparseCut.

## 4 EXPERIMENT

### 4.1 IMPLEMENTATION AND EXPERIMENT SETUP

**Implementation.** By default, we build atop the LLaVA-1.5 (Liu et al., 2024a) training paradigm, including model architecture, datasets, and overall training procedures. The model architecture details are described as follows. **1) Vision Encoder:** We adopt CLIP-ViT-L (Radford et al., 2021) as

Table 1: Results of representative MLLMs on academic-task-oriented benchmarks. Configurations include the LLM backbone, effective image resolution, number of visual tokens, visual context length, and pretraining (PT) / supervised fine-tuning (SFT) data sizes. Most methods use visual tokens as input embeddings, making the number of visual tokens equal to the visual context length. DeepStack-L and SparseCut additionally incorporate high-resolution visual tokens, resulting in a 4× longer visual context. Asterisks ("*") indicate results reproduced from the DeepStack (Meng et al., 2024) paper. *Eff.Res.* means Efficient Resolution. *Vis.Len.* means Visual tokens length. *Con.Len.* means Context Length.

| Method | Configuration | | | | | | Benchmark | | | | |
| | LLM | Eff. Res. | Vis. Len. | Con. Len. | PT | SFT | VQAv2 | GQA | VizWiz | SciQA -IMG | TextVQA |
| --- | --- | --- | --- | --- | --- | --- | --- | --- | --- | --- | --- |
| BLIP-2 | Vicuna-13B | 224 | 32 | 32 | 129M | - | 65.0* | 41.0* | 19.6* | 61.0* | 42.5* |
| InstructBLIP | Vicuna-7B | 224 | 32 | 32 | 129M | 1.2M | - | 49.2* | 34.5* | 60.5* | 50.1* |
| InstructBLIP | Vicuna-13B | 224 | 32 | 32 | 129M | 1.2M | - | 49.5 | 33.4 | 63.1 | 50.7 |
| Shikra | Vicuna-13B | 224 | - | - | 600K | 5.5M | 77.4* | - | - | - | - |
| IDEFICS-9B | LLaMA-7B | 224 | - | - | 353M | 1M | 50.9* | 38.4* | 35.5* | - | 25.9* |
| IDEFICS-80B | LLaMA-65B | 224 | - | - | 353M | 1M | 60.0* | 45.2* | 36.0* | - | 30.9* |
| Qwen-VL | Qwen-7B | 448 | 256 | 256 | 1.4B | 50M | 78.8* | 59.3* | 35.2* | 67.1* | 63.8* |
| LLaVA | Vicuna-7B | 336 | 576 | 576 | 558K | 665K | 78.5* | 62.0* | 50.0* | 66.8* | 58.2* |
| DeepStack-L | Vicuna-7B | 672 | 2880 | 576 | 558K | 665K | 79.5* | 63.1* | 50.3 | 67.2 | **62.4*** |
| SparseCut | Vicuna-7B | 672 | 2880 | 576 | 558K | 665K | **80.0** | **63.6** | **54.2** | **70.0** | 60.9 |
| LLaVA | Vicuna-13B | 336 | 576 | 576 | 558K | 665K | 80.0* | 63.3* | 53.6* | 71.6* | 61.3* |
| SparseCut | Vicuna-13B | 672 | 2880 | 576 | 558K | 665K | **81.8** | **64.0** | **56.1** | **71.8** | **61.9** |

the visual encoder, consistent with LLaVA. **2) LLM:** We experiment with language models ranging from 3B to 13B parameters. The main experiments use Vicuna-7B and Vicuna-13B, both fine-tuned from LLaMA2 using the LLaVA configuration. **3) Modality Adapter:** Implemented as a single-layer Transformer block with randomly initialized weights, the adapter uses self-attention with a hidden size matching the ViT encoder and a feedforward module identical to that in LLaVA. By default, the number of shortcut connections is eight, with all connection ends spreading across the ViT and LLM in uniform intervals: three for ViT with 24 layers, four for Vicuna-7B with 32 layers, and five for Vicuna with 40 layers.

**Training Datasets.** To ensure a fair comparison, we use the same datasets as LLaVA-1.5 (Liu et al., 2024a). The pretraining set contains approximately *558K* image-caption pairs, and the fine-tuning set includes *665K* multi-turn dialogues.

**Training Recipe.** Experiments are conducted on a server with four NVIDIA H200 GPUs. Training proceeds in two stages: 1) Pretraining. The ViT and LLM are initialized from pretrained checkpoints, while the modality adapter is trained from scratch. Only the adapter is updated, using a batch size of 256 and a learning rate of 1e-3. 2) Fine-tuning. The ViT remains frozen, and both the adapter and LLM are updated using a batch size of 128 and a learning rate of 2e-5. Each stage runs for one epoch to ensure consistency and comparability.

**Baselines.** We adopt a diverse set of representative baselines, including BLIP2 (Li et al., 2023b), InstructBLIP (Dai et al., 2023), Shikra (Chen et al., 2023), and IDEFICS (Laurençon et al., 2023), following the evaluation setup of LLaVA-1.5 (Liu et al., 2024a). Additionally, we include comparisons with the most related and recent state-of-the-art method, DeepStack (Meng et al., 2024). Notably, while DeepStack-V employs a trainable vision encoder, we report comparisons against DeepStack-L to maintain fair evaluation, as it adopts the same frozen-encoder setting as ours.

## 4.2 MAIN RESULTS

We evaluate SparseCut across a diverse set of benchmarks using base LLMs of different scales. Table 1 reports performance on academic-task-oriented benchmarks, including VQAv2 (Goyal et al., 2017), GQA (Hudson & Manning, 2019), VizWiz (Gurari et al., 2018), SciQA-IMG (Lu et al., 2022), and TextVQA (Singh et al., 2019). Table 2 presents results on instruction-following benchmarks, including POPE (Li et al., 2023d), MMBench (English and Chinese) (Liu et al., 2024b), and Seed-

Table 2: Results on instruction following benchmarks. In the LLM column, V denotes Vicuna, L denotes LLaMA, and Q denotes Qwen.

| Method | Configuration | | | | | | Benchmark | | | | | |
| | LLM | Eff. Res. | Vis. Len. | Con. Len. | PT | SFT | POPE | | | MMBench | | SEEDBench |
| | | | | | | | rand | pop | adv | EN | CN | |
| BLIP2-14B | V-13B | 224 | 32 | 32 | 129M | - | 89.6* | 85.5* | 80.9* | - | - | 49.7* |
| InstructBLIP | V-7B | 224 | 32 | 32 | 129M | 1.2M | - | - | - | 36* | 23.7* | 58.8* |
| InstructBLIP | V-13B | 224 | 32 | 32 | 129M | 1.2M | 87.7* | 77* | 72* | - | - | - |
| Shikra | V-13B | 224 | - | - | 600K | 5.5M | - | - | - | 58.8* | - | - |
| IDEFICS-9B | L-7B | 224 | - | - | 353M | 1M | - | - | - | 48.2* | 25.2* | 44.5* |
| IDEFICS-80B | L-65B | 224 | - | - | 353M | 1M | - | - | - | 54.5* | * 39.1* | 53.2* |
| Qwen-VL | Q-7B | 448 | 256 | 256 | 1.4B | 50M | - | - | - | 38.2* | 7.4* | 62.3* |
| LLaVA | V-7B | 336 | 576 | 576 | 558K | 665K | 87.3* | 86.1* | 84.2* | 64.3* | 58.3* | 66.1* |
| DeepStack-L | V-7B | 672 | 2880 | 576 | 558K | 665K | 86.7 | 86.9 | 84.9 | 65.2 | 54.8 | 65.8 |
| SparseCut | V-7B | 672 | 2880 | 576 | 558K | 665K | **89.0** | **88.0** | **86.0** | **67.3** | **60.1** | **67.3** |
| LLaVA | V-13B | 336 | 576 | 576 | 558K | 665K | 87.1* | 86.2* | 84.5* | 67.7* | 63.6* | 68.2* |
| SparseCut | V-13B | 672 | 2880 | 576 | 558K | 665K | **89.4** | **87.8** | **86.1** | **69.7** | **64.1** | **69.0** |

Table 3: Results on different shortcut connection patterns corresponding to Fig. 2. In the methods listed in the first column, D denotes Dense, S denotes Sparse, B denotes Bottom-skewed, T denotes Top-skewed, U denotes Uniform, and RU denotes Reverse Uniform.

| | VQAv2 | GQA | VizWiz | SciQA-IMG | TextVQA | MMBench-EN | MMBench-CN | SEEDBench-IMG | **Avg.** |
| --- | --- | --- | --- | --- | --- | --- | --- | --- | --- |
| LLaVA | 78.5 | 62.0 | 50.0 | 66.8 | 58.2 | 64.3 | 58.3 | 66.1 | 63.0 |
| (D+B) (Fig. 2(a)) | 79.0 | 62.2 | 50.3 | 68.7 | 58.4 | 65.6 | 59.0 | 66.3 | 63.7 |
| (D+U) (Fig. 2(b)) | 79.1 | **62.8** | 50.5 | 68.0 | 59.0 | 65.5 | 59.2 | 66.4 | 63.8 |
| (S+B) (Fig. 2(d)) | 79.2 | 62.1 | 50.7 | 68.4 | **59.4** | 65.7 | 59.0 | 66.1 | 63.8 |
| (S+T)(Fig. 2(e)) | 40.1 | 37.4 | 30.4 | 42.7 | 28.7 | 31.8 | 23.9 | 38.5 | 34.2 |
| (S+RU) (Fig. 2(f)) | 72.8 | 58.3 | 49.7 | 63.4 | 52.5 | 59.1 | 52.1 | 46.3 | 56.8 |
| (S+U) (Fig. 2(c)) | **79.9** | 62.5 | **51.7** | **69.2** | 58.6 | **66.5** | **59.8** | **66.6** | **64.4** |

Bench (Li et al., 2023a). All evaluations are conducted using the OpenCompass VLMEvalKit (Duan et al., 2024) to ensure standardized and reproducible comparisons.

The results show that SparseCut generally outperforms baseline methods using the same base LLM under comparable settings of model size and training data. On the 7B model scale, SparseCut achieves improvements ranging from 1.2 to 4.2 percentage points over LLaVA and ranging from 0.3 to 5.3 percentage points over DeepStack-L across various benchmarks, resulting in average improvements of 2.2 percentage points and 1.8 percentage points, respectively. The average improvement over LLaVA is 1.3 percentage points on the 13B scale. Note that compared to LLaVA, the performance improvement of SparseCut is achieved merely by the multi-level multi-grained shortcut mechanism without additional training data.

In particular, it shows notable gains on the VizWiz benchmark, especially for unanswerable cases—demonstrating that multi-level and multi-granular visual integration effectively mitigates hallucinations from incomplete or ambiguous inputs. SparseCut also achieves better results on both English and Chinese versions of MMBench (Liu et al., 2024b), which comprehensively evaluate visual reasoning and language understanding. These results highlight the effectiveness of our structural fusion strategy and its strong generalization across languages and question types.

## 4.3 ABLATION STUDY

### 4.3.1 EFFECTS OF VARIOUS SHORTCUT CONNECTION PATTERNS.

Table 3 presents SparseCut's performance with various representative shortcut connection patterns, aimed at analyzing how different multi-level visual integration schemes influence model perfor-

Table 4: Performance of SparseCut with and without high-resolution image input.

| | VQAv2 | GQA | VizWiz | SciQA -IMG | TextVQA | MMBench -EN | MMBench -CN | SEEDBench -IMG | Avg. |
|---|---|---|---|---|---|---|---|---|---|
| LLaVA w/o h | 78.5 | 62.0 | 50.0 | 66.8 | 58.2 | 64.3 | 58.3 | 66.1 | 63.0 |
| SparseCut w/o h | 79.9 | 62.5 | 51.7 | 69.2 | 58.6 | 66.5 | 59.8 | 66.6 | 64.4 |
| SparseCut w/ h | **80.0** | **63.6** | **54.2** | **69.9** | **60.9** | **67.3** | **60.1** | **67.3** | **65.4** |

mance. To isolate the effect of connection patterns, all experiments use single-granularity visual features without high-resolution inputs. The configurations are as follows: **1) Dense-Bottom:** Different from Dense-Uniform, this configuration concentrates the 24 connections on the lower LLM layers (bottom-skewed), as illustrated in Fig. 2(a). **2) Dense-Uniform:** This configuration uses 24 connections starting from every ViT layer and ending by evenly distributed across the 32-layer Vicuna-7B (LLM), as illustrated in Fig. 2(b). **3) Sparse-Uniform:** This is the default configuration for the main experiments. Shortcuts include eight connections with ends evenly distributed across the 24-layer ViT and 32-layer Vicuna-7B (LLM), with shallower (and deeper) ViT layers connecting deeper (and shallower) LLM layers, following the illustration in Fig. 2(c). **4) Sparse-Bottom:** A combination of the the sparse (eight connections) and bottom-skewed settings (Fig. 2(d)). **5) Sparse-Top:** Eight shortcut connections distributed evenly across ViT layers but concentrated on the top-8 LLM layers, as illustrated in Fig. 2(e). **6) Sparse-ReverseUniform:** Eight connections evenly distributed across the 24-layer ViT and 32-layer Vicuna-7B (LLM), with shallower (and deeper) ViT layers connecting shallower (and deeper) LLM layers, following the illustration in Fig. 2(f).

As shown in Table 3, generally, the uniform setting outperforms the skewed one, while the sparse setting outperforms the dense one. 1) By comparing Sparse-ReverseUniform with Sparse-Uniform, we can infer that the U-shape connection order is superior to the aligned-depth order, suggesting that the hierarchical fusion of visual and textual tokens reduces representation gaps. This finding aligns with the design intuition of the U-Net architecture: shallow visual features benefit from limited fusion, whereas deeper semantic features require broader integration to support high-level cross-modal reasoning. 2) By comparing Dense-Uniform with Sparse-Uniform, we can infer that while dense connections suffer from potential redundancy and interference, sparse shortcuts can reduce redundant information flow and improve information exchange efficiency between the vision encoder and LLM. 3) By comparing Sparse-Bottom, Sparse-Top, and Sparse-Uniform, the uniform pattern beats the skewed patterns, suggesting that balanced vision-language fusion across layer depth can better handle semantic mismatch. Specifically, Sparse-Top shows a clear performance drop, by around half compared to Sparse-Uniform, confirming the intuition that bypassing most LLM decoder layers prevents the model from leveraging its full hierarchical reasoning capacity. Sparse-Uniform performs better in most tasks than other patterns. These findings provide a generally preferred design choice of the three factors forming the SparseCut framework: **U-shape connection order, uniform connection-end distribution, and sparse density**.

### 4.3.2 EFFECTS OF MULTI-GRAINED FUSION OF VISUAL FEATURES.

To assess the impact of the proposed high- and low-resolution visual fusion strategy, we conducted ablation experiments under identical model settings: 1) using only low-resolution features, and 2) combining both high- and low-resolution features. As shown in Table 4, the inclusion of high-resolution features leads to consistent performance improvements across all benchmarks. This suggests that high-resolution inputs provide complementary fine-grained visual details that are otherwise absent in low-resolution representations. These details are especially critical for tasks requiring precise object recognition or spatial reasoning, such as VizWiz and SciQA-IMG. SparseCut integrates high-resolution information efficiently via sparse shortcut connections, avoiding the quadratic cost associated with full-resolution token expansion in traditional multimodal LLMs. This allows the model to capture richer semantics without compromising scalability.

### 4.3.3 ALTERNATIVE BASE LLMS

To assess the generalizability of SparseCut, we replace the original Vicuna (Zheng et al., 2023) backbone with the 36-layer Qwen2.5-3B (Team, 2024). We apply Sparse-Uniform method with

Table 5: Performance of alternative base LLMs on benchmarks without requiring online submission. Symbol ‡ indicates results reported on validation sets. † denotes our SparseCut (S+U) method.

| | GQA | VizWiz | SCIQA-IMG | TextVQA | POPE$_{avg}$ | MMEN | MMCN | SEEDBench |
|---|---|---|---|---|---|---|---|---|
| Qwen | 61.1 | 40.3‡ | 52.9 | 51.7 | 81.1 | 56.6 | 58.3 | 60.9 |
| Qwen w/ † | 62.0 | 41.1‡ | 53.8 | 53.0 | 82.0 | 59.0 | 58.7 | 62.2 |

Table 6: SparseCut's stability when training with various data volumes and unfrozen ViT. ‡ represents the result from validation. Since the online submission portal for part of the VizWiz test set has been closed, we report results on the validation set instead.

| | GQA | VizWiz | SciQA-IMG | TextVQA | POPE$_{avg}$ | MMEN | MMCN | SEEDBench |
|---|---|---|---|---|---|---|---|---|
| 100%PT+20%SFT | 57.4 | 50.3 ‡ | 67.1 | 53.9 | 85.9 | 59.0 | 53.7 | 61.2 |
| 100%PT+40%SFT | 60.6 | 52.0 ‡ | 68.3 | 57.3 | 86.2 | 60.1 | 52.4 | 62.0 |
| 100%PT+60%SFT | 61.7 | 52.2 ‡ | 68.3 | 58.7 | 86.5 | 64.8 | 57.6 | 61.9 |
| 100%PT+80%SFT | 63.1 | 53.5 ‡ | 69.1 | 59.8 | 87.0 | 65.7 | 59.4 | 64.6 |
| 50%PT+100%SFT | 63.4 | 53.3 ‡ | 69.3 | 60.0 | 87.2 | 66.6 | 58.6 | 67.0 |
| **100%PT+100%SFT** | **63.6** | **54.2** | **70.0** | **60.9** | **87.6** | **67.3** | **60.1** | **67.3** |
| **100%PT+100%SFT Unfrozen ViT** | **65.7** | **57.4** ‡ | **71.9** | **62.0** | **88.0** | **68.9** | **62.1** | **68.4** |

six shortcut connections to match its deeper architecture, while keeping all other experimental settings unchanged. This evaluation isolates the effect of the base LLM on performance and tests the method's adaptability across architectures. As shown in Table 5, our method consistently yields notable improvements across all tasks, demonstrating robustness across LLM backbones. These results further indicate that SparseCut is not dependent on a specific LLM design, but instead offers a broadly applicable solution for enhancing cross-modal understanding across diverse model configurations.

### 4.3.4 TRAINING STABILITY: VARYING TRAINING DATA SIZE AND UNFREEZING VIT

We explore the training stability of SparseCut with various sizes of pretraining and supervised fine-tuning (SFT) data, as well as unfrozen ViT. In the setting with low- and high-resolution image inputs, we first keep ViT frozen during training and vary the pretraining and SFT data size. Since the pretraining phase only updates the Adapter module, we evaluate two pretraining data settings: 50% and 100%. SFT data are proportioned into five subsets: 20%, 40%, 60%, 80% and 100%. As shown in Table 6, increasing the amount of data in the SFT stage consistently leads to stronger model performance, exhibiting a steady upward trend. In contrast, the pretraining stage has a relatively smaller impact on the final results, since it only trains the alignment module, further indicating that our alignment module is already sufficiently trained. We then unfreeze the ViT and train the model with full pretraining and SFT data. As shown in Table 6, training the ViT along with the LLM consistently improves SparseCut across all tasks. It indicates that the structural connections between the ViT and LLM enable the visual features better aligned in semantics with the textual features during training. These results validate SparseCut's training stability with varying training data sizes and unfrozen ViT, which is usually fine-tuned for domain-specific downstream tasks.

## 5 CONCLUSION

This paper presents SparseCut, a general cross-modal fusion architecture for MLLMs. By introducing sparse shortcut connections and joint high-/low-resolution modeling, SparseCut achieves efficient multi-level visual integration without inflating context length or computational cost. Experiments across diverse benchmarks demonstrate its effectiveness and the potential application of different shortcut patterns to vision-language model training.

## REPRODUCIBILITY STATEMENT

The source code for reproduction is available in the supplementary material. The required datasets can be obtained from a third-party repository by following the instructions. The test of some benchmarks requires online submission with Internet access.

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

Table 7: Additional baseline comparisons, including Qwen2.5-VL and LLaVA-Next.

| | GQA | VizWiz | SCIQA | TextVQA | POPE | | | MM | | SEEDBench |
| | | | | | rand | pop | adv | EN | CN | |
|---|---|---|---|---|---|---|---|---|---|---|
| Qwen2.5-VL | 60.3 | 41.4 | 73.0 | 84.7 | 88.5 | 87.5 | 86.4 | 83.0 | 82.5 | 76.5 |
| LLaVA-Next | 63.0 | 49.7 | 68.3 | 63.6 | 88.1 | 86.8 | 85.3 | 67.8 | 59.5 | 69.0 |
| SparseCut | 63.4 | 54.2 | 70.0 | 60.9 | 89.0 | 88.0 | 86.0 | 67.3 | 60.1 | 67.3 |
| SparseCut with unfrozen VIT | 65.7 | 57.4 | 71.9 | 62.0 | 89.0 | 88.3 | 86.7 | 68.9 | 62.1 | 68.4 |

# APPENDIX

## A SOURCE CODE

The source code is available in the supplementary material.

## B THE USE OF LARGE LANGUAGE MODELS (LLM)

In preparing this manuscript, LLM was used to refine the abstract and check for grammatical issues in this paper. It was not involved in any aspect of research conception, algorithm design, or coding. All scientific contributions, including technical ideas, methodology, and experimental design, were independently developed by the authors. We assume full responsibility for the content of this work.

## C CASE STUDY

In Fig. 4, we present several examples from the VizWiz Benchmark. All samples share a common prompt: "When the provided information is insufficient, respond with 'Unanswerable'. Answer the question using a single word or phrase." For clarity, we do not show this prompt within the figures. In Fig. 4(a), LLaVA overlooks the decimal point in the reading. In Fig. 4(b), there is no clear visual evidence indicating that the image depicts a wall. In Fig. 4(c), LLaVA's response is unrelated to the question, whereas SparseCut outputs Unanswerable because the question does not specify any identifiable object in the image. In Fig. 4(d), "jala" is merely a fragment of text appearing on the object and does not indicate what the object actually is. In Fig. 4(e), LLaVA incorrectly interprets a blurry coin as the sun. In Fig. 4(f), the brand of gum is indiscernible from the image, yet LLaVA answers "doublemint." In Fig. 4(g), LLaVA overlooks the critical word "pork" shown on the can. In Fig. 4(h), the question asks for the expiration date, which cannot be determined from the image; however, LLaVA hallucinates "17 02 2020" likely due to the presence of "17 oz" text.

## D MORE BASELINE

We report the performance of our method on several benchmarks using the Qwen2.5-VL and LLaVA-Next in Table 7. While our approach achieves comparable or even slightly better results on certain benchmarks, we also acknowledge that it lags significantly behind on others.

However, it is important to note that Qwen2.5-VL is an industrial-scale model. According to its technical report, it is trained on approximately 4T tokens, whereas our training corpus contains only about 704M tokens based on our statistics (Qwen2.5-VL's training tokens are approximately 5,000 times more than ours). According to the LLaVA-Next blog, its pre-training data scale is the same as ours, but its SFT stage uses substantially more data samples, approximately 100K more, and additionally unfreezes the ViT during training. Therefore, its results should be compared against our experiments where the ViT is unfrozen. Though we claim that the advantage of SparseCut should be compared based on the base model it is integrated with (LLaVA in our default settings), SparseCut surprisingly outperforms these heavily trained VL models in quite a few tasks, such as GQA, VizWiz, and POPE.

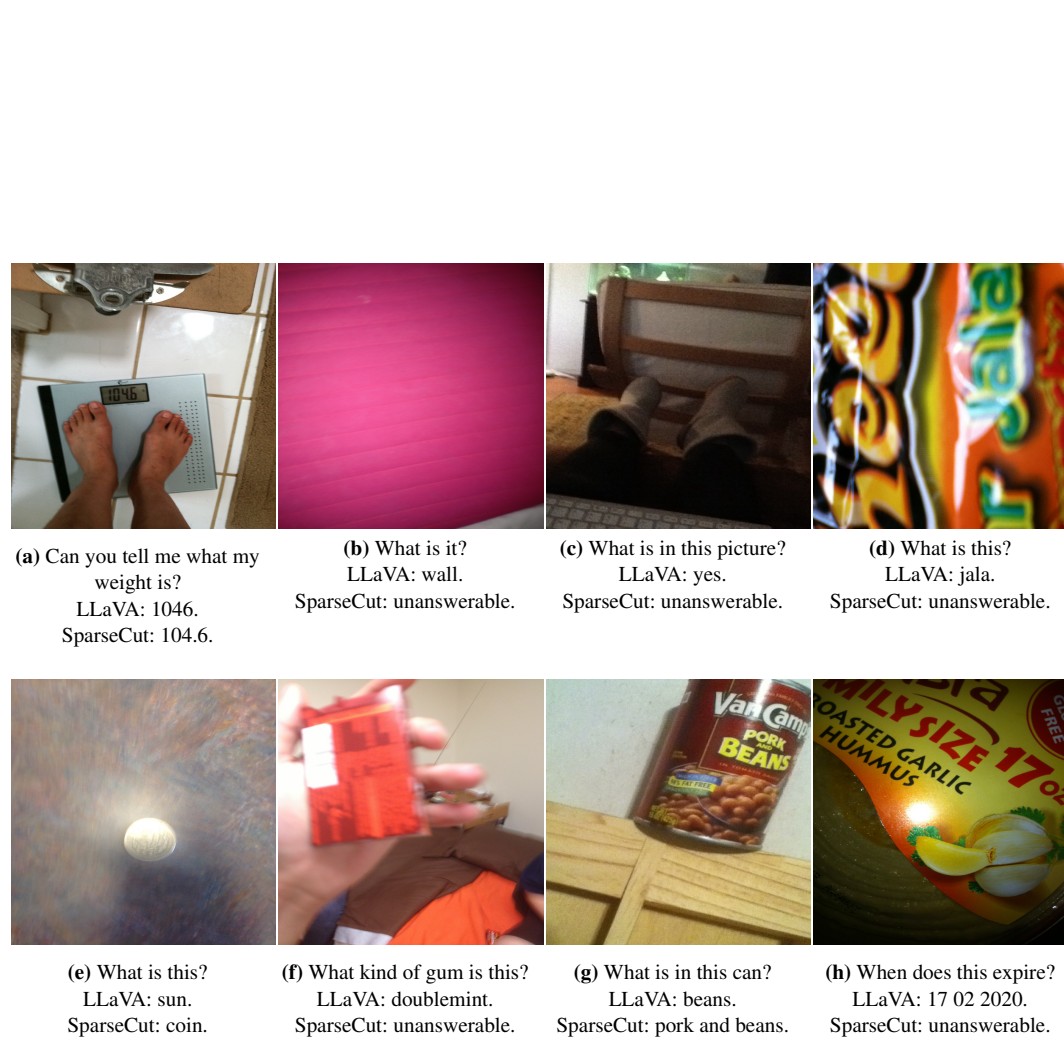

Figure 4: Case study examples on the VizWiz benchmark.

