# OpenReview forum: "Sparse Shortcuts: Facilitating Efficient Fusion in Multimodal Large Language Models"
_ICLR.cc/2026/Conference — Submitted to ICLR 2026_

### Official Review · Reviewer_UcZ1 · 2025-10-30

**Soundness:** 3
**Presentation:** 2
**Contribution:** 3
**Rating:** 4
**Confidence:** 3

**Summary:**

This paper proposes SparseCut, a cross-modal fusion architecture for multimodal large language models (MLLMs). The key idea is to introduce sparse shortcut connections between layers of the vision encoder (ViT) and the language model (LLM). SparseCut proposes an efficient multi-grained fusion module, which merges low- and high-resolution image features via cross-attention within each shortcut before they are fed into the LLM. This allows multi-resolution integration without increasing the token sequence length, thereby avoiding quadratic attention cost. Experiments on several benchmarks (VQAv2, GQA, VizWiz, SciQA-IMG, MMBench, SEEDBench, etc.) show consistent improvements over LLaVA-1.5 and DeepStack, both on Vicuna-7B and Vicuna-13B backbones.

**Strengths:**

- The paper’s motivation and contributions are explicitly stated and consistently reflected in the methodology and experiments.
- The multi-grained fusion approach seems to address the inefficiency of concatenating high-resolution tokens, offering a simple architectural solution.

**Weaknesses:**

- The baseline (DeepStack [1]) shown in the tables of the manuscript appears to be relatively outdated, dating back to June 2024. Could the authors provide comparisons with more recent models or benchmarks?
- Some baselines (e.g., Qwen2.5-VL, VITRON, LLaVA-NeXT) are missing, making the claimed SOTA improvement less convincing.
- It is unclear whether this mechanism can be well scaled to video or higher resolution inputs, which are more important for MLLM processing long sequences.

[1] Meng, Lingchen, et al. "Deepstack: Deeply stacking visual tokens is surprisingly simple and effective for lmms." Advances in Neural Information Processing Systems 37 (2024): 23464-23487.

**Questions:**

Please refer to the weakness.

---

> ### Author Response · Authors · 2025-11-22
>
> Thank you for your constructive feedback. Regarding your questions and suggestions,  we provide detailed clarifications below. If you have any follow-up questions or comments, please let us know, and we will be happy to discuss further.
>
> **Weakness 1: (About baseline)** The baseline (DeepStack [1]) shown in the tables of the manuscript appears to be relatively outdated, dating back to June 2024. Could the authors provide comparisons with more recent models or benchmarks?
>
> **A1:** DeepStack-VL is, in fact, the most recent closely related work that specifically targets structured visual–language information fusion, and it aligns well with our setting in terms of model configuration, data scale, and training paradigm. More recent models, such as Qwen2.5-VL, are industrial-scale systems; although they offer 7B-sized variants, their base LLMs, ViT architectures, training strategies, and training volumes differ substantially. These discrepancies introduce many uncontrolled variables, making such comparisons less meaningful for analyzing structured fusion mechanisms.
>
> **Weakness 2: (About baseline)** Some baselines (e.g., Qwen2.5-VL, VITRON, LLaVA-NeXT) are missing, making the claimed SOTA improvement less convincing.
>
> **A2:** We add the performance of our method on several benchmarks using the Qwen2.5-VL and LLaVA-Next in the appendix. While our approach achieves comparable or even slightly better results on certain benchmarks, we also acknowledge that it lags significantly behind on others.
>
> However, it is important to note that Qwen2.5-VL is an industrial-scale model. According to its technical report [1], it is trained on approximately **4T** tokens, whereas our training corpus contains only about **704M tokens** based on our statistics (Qwen2.5-VL’s training tokens are approximately 5,000 times more than ours).  According to the LLaVA-Next blog[2], its pre-training data scale is the same as ours, but its SFT stage uses substantially more data samples, approximately 100K more, and additionally unfreezes the ViT during training. Therefore, its results should be compared against our experiments where the ViT is unfrozen. Though we claim that the advantage of SparseCut should be compared based on the base model it is integrated with (LLaVA in our default settings), SparseCut surprisingly outperforms these heavily trained VL models in quite a few tasks, such as GQA, VizWiz, and POPE.
>
> |  | GQA | VizWiz | SCIQA | TextVQA | POPE | MMEN | MMCN | SEEDBench |
> | --- | --- | --- | --- | --- | --- | --- | --- | --- |
> | Qwen2.5-VL | 60.3 | 41.4 | 73.0 | 84.7 | R:88.5P:87.5A:86.4 | 83.0 | 82.5 | 76.5 |
> | LLaVA-Next | 63.0 | 49.7 | 68.3 | 63.6 | R:88.1P:86.8A:85.3 | 67.8 | 59.5 | 69.0 |
> | SparseCut | 63.4 | 54.2 | 70.0 | 60.9 | R:89.0P:88.0A:86.0 | 67.3 | 60.1 | 67.3 |
> | SparseCut with unfrozen VIT | 65.7 | 57.4 | 71.9 | 62.0 | R:89.0P:88.3A:86.7 | 68.9 | 62.1 | 68.4 |
>
> **Evaluating SparseCut's performance with a different base LLM can address the reviewer's concern about more convincing evidence for performance improvement.** Therefore, we include experiments with a Qwen2.5 as the base model to validate generality in Section 4.3.3 in the updated manuscript. Specifically, we evaluated:
>
> • CLIP-ViT + MLP + Qwen2.5-3B (LLaVA-style pipeline), and
>
> • CLIP-ViT + Adapter + Qwen2.5-3B (our SparseCut paradigm).
>
> The results in Table 5 demonstrate that our proposed paradigm remains consistently effective with Qwen2.5 as the base LLM, further convincing the effectiveness of SparseCut.
>
> |  | GQA | VizWiz | SCIQA-IMG | TextVQA | $\text{POPE}_{\text{avg}}$   | MMEN | MMCN | SEEDBench |
> | --- | --- | --- | --- | --- | --- | --- | --- | --- |
> | vanilla Qwen | 61.1 | 40.3 | 52.9 | 51.7 | 81.1 | 56.6 | 58.3 | 60.9 |
> | Qwen+SparseCut | **62.0** | **41.1** | **53.8** | **53.0** | **82.0** | **59.0** | **58.7** | **62.2** |
>
> [1]  Bai S, Chen K, Liu X, et al. Qwen2. 5-vl technical report[J]. arXiv preprint arXiv:2502.13923, 2025.
>
> [2]  LiuHaotian, LiChunyuan, LiYuheng, LiBo, ZhangYuanhan, ShenSheng, LeeYong Jae. LLaVA-NeXT: Improved reasoning, OCR, and world knowledge , 2024.

---

> > ### Author Response · Authors · 2025-11-23
> >
> > **Weakness 3: (About the adaptability)** It is unclear whether this mechanism can be well scaled to video or higher resolution inputs, which are more important for MLLM processing long sequences.
> >
> > **A3:** This question naturally inspires promising future research directions with the SparseCut framework. We focus the discussion on the adaptability to video and higher-resolution inputs from the perspectives of engineering effort and potential intuitive advantages.
> >
> > Current large-scale video understanding models predominantly follow a “**frame sampling + token concatenation**” paradigm. Under this mainstream pipeline, our mechanism naturally extends to video. **Each frame can adopt the same hierarchical feature extraction and structured fusion strategy as in the image case**. Our high-resolution handling strategy also carries over to high-resolution video, though the extremely long token sequences caused by frame-wise concatenation are indeed a challenge that we have not addressed and represent a broader open problem in the field.
> >
> > For higher resolution images, our method remains broadly applicable. The two key components of SparseCut, which are **(1) leveraging multi-level visual features and (2) reducing the effective token length through structured injection paths, are conceptually resolution-agnostic.** High-resolution ViTs, pyramid-style encoders, or patch merging strategies can all be naturally integrated with SparseCut.
> >
> > Overall, with predictable engineering adaptations, the structured information fusion design of SparseCut offers high potential advantages for applying to videos and higher-resolution inputs. We deem this an inspiration for a range of meaningful future research on structurally fusing vision and language features for broader applications.

---

### Official Review · Reviewer_opN4 · 2025-10-31

**Soundness:** 3
**Presentation:** 4
**Contribution:** 3
**Rating:** 6
**Confidence:** 3

**Summary:**

This paper introduces SparseCut, a general multimodal fusion framework for MLLMs, which enhances cross-modal understanding by establishing sparse shortcut connections between multiple layers of the vision encoder and the language model. These shortcuts allow hierarchical and multi-grained visual information to be integrated efficiently without extending the LLM’s input length. By fusing high- and low-resolution visual features through cross-attention, SparseCut preserves rich semantics while maintaining computational efficiency. Experiments on various benchmarks show consistent improvements over LLaVA and DeepStack, achieving better performance with minimal additional cost and strong scalability across different base LLMs.

**Strengths:**

1. This paper proposes a method that can efficiently integrates multi-level and multi-resolution visual features without increasing computational cost.
2. Through shortcut connections, SparseCut effectively incorporates multi-granularity visual features into the LLM while preserving its original context length and computational efficiency.
3. The experimental results demonstrate strong generalization and scalability across different base LLMs.

**Weaknesses:**

1. The choice of shortcut pattern (density, distribution) may require manual tuning.
2. The method relies on a frozen vision encoder, potentially limiting deeper cross-modal alignment.

**Questions:**

1. The choice of shortcut pattern (density, distribution) may require manual tuning.
2. The method relies on a frozen vision encoder, potentially limiting deeper cross-modal alignment.

---

> ### Author Response · Authors · 2025-11-22
>
> Thank you for your constructive feedback. Regarding your questions and suggestions,  we provide detailed clarifications below. If you have any follow-up questions or comments, please let us know, and we will be happy to discuss further.
>
> **Weakness 1:** The choice of shortcut pattern (density, distribution) may require manual tuning.
>
> **A1:** We claim that our main purpose is to uncover general design choices of the three decisive factors forming the SparseCut framework: connection order, connection-end distribution, and density. Therefore, in the updated manuscript, we have discussed in more detail the mechanism of different settings of these factors in Section 3.2 and compared their performance in Section 4.3.1. By exploring the combination of different settings of the three factors, which form a number of representative shortcut patterns, **we can propose generally preferred design choices for these factors: U-shape connection order, uniform connection-end distribution, and sparse density.** Guided by these findings, the manual tuning burden can be limited: discovering how sparse the density should be. Our experiment setup provides an example for setting the density: a common divisor of the numbers of ViT layers and LLM layers (8 connections for a 24-layer ViT and 32-layer LLM).
>
> **Weakness 2:** The method relies on a frozen vision encoder, potentially limiting deeper cross-modal alignment.
>
> **A2:** About the reliance on a frozen vision encoder, we acknowledge that freezing the vision encoder may limit the upper bound of model performance. However, we adopted this strategy in the main experiments to remain consistent with baseline settings and to ensure a fair and controlled comparison. In response to the reviewer’s suggestion, we additionally report results with the vision encoder unfrozen in Section 4.3.4. Under the same amount of training data, allowing the vision encoder to update further improves the model’s performance ceiling across all tasks. It indicates that the structural connections between the ViT and LLM enable the visual features better aligned in semantics with the textual features during training. It also validates SparseCut's training stability with unfrozen ViT, which is usually fine-tuned for domain-specific downstream tasks.
>
> |  | GQA | VizWiz | SciQA-IMG | TextVQA | $\text{POPE}_{\text{avg}}$  | MMEN | MMCN | SEEDBench |
> | --- | --- | --- | --- | --- | --- | --- | --- | --- |
> | SparseCut | 63.6 | 54.2 | 70.0 | 60.9 | 87.6 | 67.3 | 60.1 | 67.3 |
> | SparseCut with Unfrozen ViT | **65.7** | **57.4**  | **71.9** | **62.0** | **88.0** | **68.9** | **62.1** | **68.4** |

---

### Official Review · Reviewer_PBEk · 2025-10-31

**Soundness:** 2
**Presentation:** 2
**Contribution:** 2
**Rating:** 4
**Confidence:** 4

**Summary:**

This work presents SparseCut, a cross-modal fusion approach that introduces sparse shortcut pathways to efficiently inject multi-level visual information into LLMs. The proposed design aims to enhance semantic fusion while maintaining low computational overhead for the LLM. Experimental results are reported to show notable performance gains across several benchmarks, suggesting promising generalization.

**Strengths:**

1. The overall idea is conceptually clear and well motivated.
2. The evaluation covers a reasonably broad set of benchmarks, demonstrating the generalization capabilities of the approach.
3. The manuscript is clearly written and easy to read.

**Weaknesses:**

1. The experiments are confined to the Vicuna-based LLaVA framework. To support the claim of wide applicability, additional validation on more diverse and up-to-date LLM backbones (e.g., Qwen2.5 series) and multimodal architectures (e.g., Qwen2.5-VL, InternVL-2.5) would be essential.
2. The paper emphasizes efficiency on the language side but neglects the additional cost incurred by processing higher-resolution images through the vision encoder. Reporting overall metrics such as end-to-end FLOPs or inference FPS would provide a more accurate assessment of the actual computational burden.
3. The baseline (LLaVA-1.5) is evaluated at a lower image resolution, whereas the proposed method adopts a much higher resolution. Since increased resolution itself can yield substantial performance improvement, this discrepancy makes it difficult to isolate the contribution of the SparseCut design. A fair comparison under identical resolution settings is needed.

**Questions:**

See Weakness

---

> ### Author Response · Authors · 2025-11-22
>
> Thank you for your constructive feedback. Regarding your questions and suggestions,  we provide detailed clarifications below. If you have any follow-up questions or comments, please let us know, and we will be happy to discuss further.
>
> **Weakness 1: (About different and up-to-date LLM backbones and multimodal architectures)** The experiments are confined to the Vicuna-based LLaVA framework. To support the claim of wide applicability, additional validation on more diverse and up-to-date LLM backbones (e.g., Qwen2.5 series) and multimodal architectures (e.g., Qwen2.5-VL, InternVL-2.5) would be essential.
>
> **A1:** Qwen2.5-VL is a fully trained multimodal large model with its own carefully designed alignment and modeling strategy, learned from massive multimodal data. Applying our method to such a model and fine-tuning it with only a small amount of data would be inappropriate: the model would not have enough data to learn our new integration strategy, and simultaneously it would violate the train–inference consistency of its original modeling paradigm. Therefore, Qwen2.5-VL is not an ideal testbed for verifying architectural generality under limited-data settings.
>
> Beyond this consideration, we did include experiments with a different LLM backbone to validate generality in the updated manuscript. Specifically, we evaluated:
>
> • CLIP-ViT + MLP + Qwen2.5-3B (LLaVA-style pipeline), and
>
> • CLIP-ViT + Adapter + Qwen2.5-3B (our SparseCut paradigm).
>
> The results demonstrate that our proposed paradigm remains effective consistently across different LLM backbones, further supporting its broad applicability.
>
> |  | GQA | VizWiz | SCIQA-IMG | TextVQA | $\text{POPE}_{\text{avg}}$   | MMEN | MMCN | SEEDBench |
> | --- | --- | --- | --- | --- | --- | --- | --- | --- |
> | Qwen | 61.1 | 40.3 | 52.9 | 51.7 | 81.1 | 56.6 | 58.3 | 60.9 |
> | Qwen w/ shortcut | 62.0 | 41.1 | 53.8 | 53.0 | 82.0 | 59.0 | 58.7 | 62.2 |
>
> **Weakness 2: (About computational efficiency)** The paper emphasizes efficiency on the language side but neglects the additional cost incurred by processing higher-resolution images through the vision encoder. Reporting overall metrics such as end-to-end FLOPs or inference FPS would provide a more accurate assessment of the actual computational burden.
>
> **A2:** We have compared the FLOPs of LLvVA and SparseCut in low-resolution and hybrid-resolution settings in Section 3.3 of the undated manuscript. The ViT we use produces 576 image tokens after preprocessing (here we only consider image tokens and ignore the text tokens, including the question or answer). Under this setting, the FLOPs of both our ViT and LLM components are identical to those in LLaVA.
>
> In the hybrid-resolution scenarios, both our method and LLaVA require five ViT forward passes. For the LLM part, LLaVA adopts a concatenation-based strategy, resulting in an input length of 2880 vision tokens and a FLOPs cost of 41.6T. In contrast, our method performs token-length reduction, yielding an input length of 576 tokens and a corresponding FLOPs cost of 7.63T.
>
> **In summary, Low resolution:**
>
> $LLaVA: 0.38T(VIT) + 0.025T(MLP) + 7.63T (LLM) \approx 8.04T $
>
> $SparseCut: 0.38T(VIT) + 0.031T(Adapter) + 7.63T (LLM) \approx 8.04T$
>
> **Hybrid resolution:**
>
> $LLaVA: 0.38T * 5(VIT) + 0.12T(MLP) + 41.6T(LLM) \approx 43.62T $
>
> $SparseCut: 0.38T * 5(VIT) + 0.045T(MLP) + 7.63T(LLM) \approx 9.6T$
>
> **Weakness 3: (About fair comparison)** The baseline (LLaVA-1.5) is evaluated at a lower image resolution, whereas the proposed method adopts a much higher resolution. Since increased resolution itself can yield substantial performance improvement, this discrepancy makes it difficult to isolate the contribution of the SparseCut design. A fair comparison under identical resolution settings is needed.
>
> **A3:** To eliminate the influence of resolution differences, we added a controlled comparison in Table 4. Specifically, **SparseCut w/o h** denotes the setting where our method uses exactly the same 336×336 input resolution as LLaVA during both training and evaluation. Even under this identical-resolution setup, SparseCut still outperforms LLaVA across all tasks by a clear margin. This demonstrates that the performance gains stem from our structured information fusion design itself, rather than from resolution advantages.
>
> |  | VQAv2 | GQA | VizWiz | SciQA-IMG | TextVQA | MMBench-EN | MMBench-CN | SEEDBench-IMG | Avg. |
> | --- | --- | --- | --- | --- | --- | --- | --- | --- | --- |
> | LLaVA | 78.5 | 62.0 | 50.0 | 66.8 | 58.2 | 64.3 | 58.3 | 66.1 | 63.0 |
> | SparseCut w/o h | 79.9 | 62.5 | 51.7 | 69.2 | 58.6 | 66.5 | 59.8 | 66.6 | 64.4 |
> | SparseCut w/ h | 80.0 | 63.6 | 54.2 | 69.9 | 60.9 | 67.3 | 60.1 | 67.3 | 65.4 |

---

### Official Review · Reviewer_YYJc · 2025-10-31

**Soundness:** 3
**Presentation:** 2
**Contribution:** 3
**Rating:** 6
**Confidence:** 4

**Summary:**

This paper proposes SparseCut, a general cross-modal fusion architecture for Multimodal Large Language Models (MLLMs) to address limitations in existing MLLMs—specifically the neglect of mid/low-level visual semantics and high computational costs from multi-grained feature integration.
Experiments validate SparseCut’s effectiveness across multiple benchmarks. Ablation studies confirm that sparse/uniform shortcuts and multi-grained fusion contribute to performance gains

**Strengths:**

1.	Addresses two critical pain points of existing MLLMs—loss of mid/low-level visual semantics (by leveraging multi-level vision encoder layers) and high computation from multi-resolution features (by fusing features before shortcut injection)—filling gaps in current cross-modal fusion designs.
2.	SparseCut is compatible with diverse base LLMs (Vicuna, Phi-3) and scales across model sizes (3.5B–13B). The shortcut pattern (order/distribution/density) is configurable, making it a flexible framework rather than a task-specific solution.
3.	By avoiding input context length expansion (a common issue with multi-grained fusion), SparseCut maintains low computational complexity while enhancing performance—critical for practical deployment of large MLLMs.

**Weaknesses:**

1.	While the paper tests sparse/uniform, dense/bottom patterns, it lacks a systematic exploration of why the U-shaped order is optimal (e.g., no comparison to linear/ random connection orders) or how to dynamically adjust shortcut density/distribution for different tasks (e.g., fine-grained recognition vs. coarse visual reasoning).

2.	The paper mentions freezing the vision encoder during training but provides no analysis of training stability (e.g., whether sparse shortcuts mitigate overfitting) or convergence speed compared to baselines. It also does not explore the impact of pretraining/fine-tuning data size on SparseCut’s performance.

3.	Quantitative benchmarks dominate the evaluation, but there is no qualitative analysis (e.g., case studies of VizWiz unanswerable questions or MMBench reasoning) to illustrate how multi-level/multi-grained fusion specifically improves cross-modal understanding (e.g., reducing hallucinations).

**Questions:**

Besides the weakness, I have extra questions:

Compared to UNet format connections, what’s the performance of the method that fuses shallow features of ViT with deeper features of LLM?

---

> ### Author Response · Authors · 2025-11-22
>
> Thank you for your constructive feedback. Regarding your questions and suggestions,  we provide detailed clarifications below. If you have any follow-up questions or comments, please let us know, and we will be happy to discuss further.
>
> **Weakness 1: (About different patterns)** (1) While the paper tests sparse/uniform, dense/bottom patterns, it lacks a systematic exploration of why the U-shaped order is optimal (e.g., no comparison to linear/ random connection orders) or (2) how to dynamically adjust shortcut density/distribution for different tasks (e.g., fine-grained recognition vs. coarse visual reasoning).
>
> **A1:** (1) In the updated manuscript, we have introduced and discussed three additional pattern variants to systematically study the effects of alternative connection orders in Section 3.2 and evaluated their performance in Section 4.3.1 with results shown in Table 3.
>
> In Section 3.2, in addition to the U-shape order,  we illustrated an alternative aligned-depth order (the new Fig. 2(f)). It reverses the U-shape to connect shallow visual layers with shallow language layers and deep visual layers with deep language layers. Formally, for any two shortcut connections $(i_1, j_1)$, $(i_2, j_2) \in S$, the aligned-depth order constraint is defined as: $i_i > i_2$ if and only if $j_1 > j_2$. We discussed the intuition of these two orders: The U-shape pattern encourages complementary information to flow bidirectionally across modalities. For example, low-level visual details have highways to inform high-level language reasoning. While multiple aligned skip connections of the aligned-depth order may introduce potential redundancy and offer diminishing returns.
>
> In Section 4.3.1, we compared the performance of patterns with different connection orders, as shown in the following table (full data is in Table 3 in the updated manuscript). The pattern of Sparse-ReverseUniform (S+RU, which is the linear order mentioned in the question) has eight connections evenly distributed across the 24-layer ViT and 32-layer Vicuna-7B (LLM), with shallower (and deeper) ViT layers connecting shallower (and deeper) LLM layers, following the illustration in Fig. 2(f). By comparing Sparse-ReverseUniform with Sparse-Uniform, we can infer that the U-shape connection order is superior to the aligned-depth order, suggesting that the hierarchical fusion of visual and textual tokens reduces representation gaps. This finding aligns with the design consideration of the U-Net architecture: shallow visual features benefit from limited fusion, whereas deeper semantic features require broader integration to support high-level cross-modal reasoning.
>
> |  | VQAv2 | GQA | VizWiz | SciQA -IMG | TextVQA | MMBench -EN | MMBench -CN | SEEDBench -IMG | Avg. |
> | --- | --- | --- | --- | --- | --- | --- | --- | --- | --- |
> | LLaVA | 78.5 | 62.0 | 50.0 | 66.8 | 58.2 | 64.3 | 58.3 | 66.1 | 63.0 |
> | (S+RU) | 72.8 | 58.3 | 49.7 | 63.4 | 52.5 | 59.1 | 52.1 | 46.3 | 56.8 |
> | (S+U) | **79.9** | **62.5** | **51.7** | **69.2** | **58.6** | **66.5** | **59.8** | **66.6** | **64.4** |
>
> (2) Instead of dynamically adjusting shortcut patterns for different tasks, we claim that a more effective and important task is to uncover general design choices of the three decisive factors forming the SparseCut framework: connection order, connection-end distribution, and density. Therefore, in the updated manuscript, we have discussed in more detail the mechanism of different settings of these factors in Section 3.2 and compared their performance in Section 4.3.1. By exploring the combination of different settings of the three factors, which form a number of representative shortcut patterns, **we can propose generally preferred design choices for these factors: U-shape connection order, uniform connection-end distribution, and sparse density.**

---

> ### Author Response · Authors · 2025-11-22
>
> **Weakness 2: （About the training stability and the impact of pretraining/fine-tuning data size)** The paper mentions freezing the vision encoder during training but provides no analysis of training stability (e.g., whether sparse shortcuts mitigate overfitting) or convergence speed compared to baselines.  It also does not explore the impact of pretraining/fine-tuning data size on SparseCut’s performance.
>
> **A2:** In the updated manuscript, we have explored the training stability by training SparseCut with the unfrozen vision encoder and various pretraining/fine-tuning data sizes in Section 4.3.4, with results presented in the new Table 6. We first keep ViT frozen during training and vary the pre-training and SFT data size.
> Since the pre-training phase only updates the Adapter module, we evaluate two pre-training data settings: 50% and 100%. SFT data are proportioned into five subsets: 20%, 40%, 60%, 80% and 100%. As shown in Table 6, increasing the amount of data in the SFT stage consistently leads to stronger model performance, exhibiting a steady upward trend. In contrast, the pretraining stage has a relatively smaller impact on the final results, since it only trains the alignment module, further indicating that our alignment module is already sufficiently trained. We then unfreeze the ViT and train the model with full pretraining and SFT data. As shown in Table 6, training the ViT along with the LLM consistently improves SparseCut across all tasks. It indicates that the structural connections between the ViT and LLM better align visual features in semantics with textual features during training. These results validate SparseCut's training stability with varying training data sizes and unfrozen ViT, which is usually fine-tuned for domain-specific downstream tasks.
>
> Table 6. SparseCut's stability when training with various data volumes and unfrozen ViT.
>
> |  | GQA | VizWiz | SciQA-IMG | TextVQA | $\text{POPE}_{\text{avg}}$ | MMEN | MMCN | SEEDBench |
> | --- | --- | --- | --- | --- | --- | --- | --- | --- |
> | 100%PT+20%SFT | 57.4 | 50.3 | 67.1 | 53.9 | 85.9 | 59.0 | 53.7 | 61.2 |
> | 100%PT+40%SFT | 60.6 | 52.0 | 68.3 | 57.3 | 86.2 | 60.1 | 52.4 | 62.0 |
> | 100%PT+60%SFT | 61.7 | 52.2 | 68.3 | 58.7 | 86.5 | 64.8 | 57.6 | 61.9 |
> | 100%PT+80%SFT | 63.1 | 53.5 | 69.1 | 59.8 | 87.0 | 65.7 | 59.4 | 64.6 |
> | 50%PT+100%SFT | 63.4 | 53.3 | 69.3 | 60.0 | 87.2 | 66.6 | 58.6 | 67.0 |
> | **100%PT+100%SFT** | **63.6** | **54.2** | **70.0** | **60.9** | **87.6** | **67.3** | **60.1** | **67.3** |
> | **100%PT+100%SFT  Unfrozen ViT** | **65.7** | **57.4** | **71.9** | **62.0** | **88.0** | **68.9** | **62.1** | **68.4** |
>
> **Weakness 3: (About Case study)** Quantitative benchmarks dominate the evaluation, but there is no qualitative analysis (e.g., case studies of VizWiz unanswerable questions or MMBench reasoning) to illustrate how multi-level/multi-grained fusion specifically improves cross-modal understanding (e.g., reducing hallucinations).
>
> **A3:**  We thank the reviewer for this insightful suggestion. Following the comment, we have added a set of qualitative case studies to Appendix C, including some examples from VizWiz (with unanswerable cases) and instances illustrating hallucination reduction.
> These case studies concretely demonstrate how multi-level/multi-grained fusion enhances cross-modal understanding. For example, by preventing the model from over-committing to uncertain visual cues, improving fine-grained recognition, and reducing hallucinations in ambiguous scenarios.

---

> ### Author Response · Authors · 2025-11-22
>
> **Question 1: (About different connection-end distribution)** Compared to UNet format connections, what’s the performance of the method that fuses shallow features of ViT with deeper features of LLM?
>
> **Response:**  Fusing shallow features of ViT with deeper features of LLM indicates a distinct connection-end distribution. To systematically study the effect of different connection-end distributions, we have discussed the mechanisms and effects of different distributions in Section 3.2 and compared their performance in Section 4.3.1 in the updated manuscript.
>
> In Section 3.2, we have illustrated and discussed three typical connection-end distributions: uniform (Fig. 2(c)), skewed to the LLM bottom (bottom-skewed, Fig. 2(d)), and skewed to the LLM top (top-skewed, Fig. 2(e)). The underlying mechanism is that the uniform distribution allows the language model to integrate hierarchical visual context in a balanced way, giving the LLM a higher opportunity to reconcile the semantic mismatch. The bottom-skewed distribution can maximize early-stage integration of visual information, thereby granting the language model greater flexibility in modeling foundational visual semantics. In the case of the top-skewed distribution,  all visual information is forced into only the top few LLM layers, bypassing most LLM decoder layers and preventing the model from leveraging its full hierarchical reasoning capacity. Consequently, the LLM may not adequately process or refine the visual representations, potentially resulting in substantially weaker image understanding.
>
> In Section 4.3.1,  we compared the performance of patterns with different connection-end distributions, as shown in the following table (full data is in Table 3 in the updated manuscript). The additional Sparse-Top (S+T) pattern fuses shallow features of ViT with deeper features of LLM. It has eight shortcut connections distributed evenly across ViT layers but concentrated on the top-8 LLM layers, as illustrated in Fig. 2(e). By comparing Sparse-Bottom, Sparse-Top, and Sparse-Uniform, the uniform pattern beats the skewed patterns, suggesting that balanced vision-language fusion across layer depth can better handle semantic mismatch. Specifically, Sparse-Top shows a clear performance drop by around half compared to Sparse-Uniform, confirming the discussion that bypassing most LLM decoder layers prevents the model from leveraging its full hierarchical reasoning capacity. Sparse-Uniform performs better in most tasks than other patterns.
>
> |  | VQAv2 | GQA | VizWiz | SciQA-IMG | TextVQA | MMBench-EN | MMBench-CN | SEEDBench-IMG | Avg. |
> | --- | --- | --- | --- | --- | --- | --- | --- | --- | --- |
> | LLaVA | 78.5 | 62.0 | 50.0 | 66.8 | 58.2 | 64.3 | 58.3 | 66.1 | 63.0 |
> | (S+B) | 79.2 | 62.1 | 50.7 | 68.4 | **59.4** | 65.7 | 59.0 | 66.1 | 63.8 |
> | (S+T) | 40.1 | 37.4 | 30.4 | 42.7 | 28.7 | 31.8 | 23.9 | 38.5 | 34.2 |
> | (S+U) | **79.9** | **62.5** | **51.7** | **69.2** | 58.6 | **66.5** | **59.8** | **66.6** | **64.4** |

---

### Author Response · Authors · 2025-12-02

Dear AC, SAC, and PC,

We extend our sincere gratitude for the time and effort you have dedicated to reading our responses and revisions to the reviewers. In this paper, we propose **Sparse Shortcuts** , a general MLLM cross-modal fusion architecture for facilitating efficient fusion in multimodal large language models.
We appreciate the reviewers’ recognition of our approach as clear and well-motivated, efficient in integrating multi-level visual features without increasing context length, and scalable with strong generalization across diverse LLM backbones and benchmarks.
**We have comprehensively and thoroughly addressed all concerns of all reviewers.** All the added content in the updated manuscript has been highlighted **in blue**. Below, we concisely summarize how we addressed all major concerns and the corresponding revisions made to the manuscript. (**We refer to Reviewer YYJc as R1, Reviewer PBEk as R2, Reviewer opN4 as R3, and Reviewer UcZ1 as R4**. )

# Shared Reviewer Concerns

**R1-W1&R3-W1 (Why U-shaped pattern is preferred & How to dynamically adjust):**

We added more variants of connection order, connection-end distribution, and density, discussed their pros and cons based on underlying mechanisms, and conducted new experiments comparing their performance, which strengthens the evidence that the U-shaped pattern is preferred.
Through analyzing and validating the roles of the three shortcut factors, we summarized strong, task-agnostic design principles: **U-shape order, uniform end distribution, and sparse density.** This general principle alleviates the necessity of dynamic task-specific adjustments. Experiments in Section 4.3.1 confirm that these principles deliver consistent advantages across diverse tasks.

**R2-W1&R4-W1,2(Request for testing with broader backbones(e.g. Qwen2.5 and Qwen2.5-VL) and more Baselines):**

We added experiments using **Qwen2.5 as the LLM backbone**(Table 5), which consistently confirm the effectiveness of our method.
Existing open-source MLLMs have been trained on extremely large-scale data (for example, Qwen2.5-VL uses around 4T tokens). Applying our method to such a model and fine-tuning it with only a small amount of data would be inappropriate: the model would not have enough data to learn our new integration strategy, and simultaneously it would violate the train–inference consistency of its original modeling paradigm.

We also added comparisons with Qwen2.5-VL and LLaVA-Next in appendix, both of which are trained with substantially larger datasets.
Despite their heavier training, our method achieves comparable or even superior results on several benchmarks, strengthening the validity of our approach.

# Individual Reviewer Concerns

**R1-W2 (Effect of pretraining/fine-tuning data scale):**

We extended Table 6 in the revised manuscript to include analyses across different PT and SFT data scales. The results show clear and expected positive trends with more data, further demonstrating the stability of our training pipeline.

**R1-W3 (Case studies):**

We added case studies in the revised appendix. These examples more clearly illustrate the improvements of our method in reducing hallucinations.

**R1-Q1 (Performance when shallow ViT features fuse with deep LLM layers):**

We added discussions and tests on fusing different-layer features, showing that directly injecting shallow ViT features into deep LLM layers bypasses most decoder layers and harms hierarchical reasoning.
We further included the **S+T experiment (Table 3)**, confirming that shallow-to-deep fusion leads to significantly poor performance across image-text tasks, consistent with our initial expectation.

**R2-W2 (Request for computational cost analysis):**

We added detailed end-to-end FLOPs calculations. With high-resolution inputs, our model requires only about one-fifth of LLaVA’s computation (9.6T vs. 43.62T), demonstrating efficiency.

**R2-W3 (Request of fair comparison using the same resolution settings across methods):**

We added low-resolution-only comparisons with the baseline in Table 4. These comparisons show that our performance improvements stem mainly from the structured shortcut design, rather than from higher input resolution.

**R3-W2 (Question about the influence of freezing the vision encoder on deeper cross-modal alignment):**

We added experiments where the ViT encoder is unfrozen. Table 6 shows that unfreezing ViT further improves downstream task alignment, indicating that the model can benefit from deeper cross-modal adaptation when allowed to update visual representations.

**R4-W3 (Reviewer asked about adaptability to video or higher-resolution inputs):**

We discussed compatibility with the mainstream video-understanding pipeline and with higher-resolution image encoders. Our analysis shows that SparseCut’s structured fusion extends naturally with predictable engineering adaptations and retains its conceptual advantages in both video and high-resolution scenarios.

---

### Meta-Review · Area_Chair_PoXQ · 2026-01-11

**Summary:**

This paper was reviewed by four experts in the field, receiving scores of 4, 4, 6, and 6. The consensus among the reviewers is that the manuscript falls below the publication threshold in its current form and requires significant revision regarding experimental evaluation, methodology, and presentation. Multiple reviewers (YYJc, PBEk, UcZ1) noted deficiencies in the evaluation. Specifically, there is a need for more rigorous ablation studies, testing across a wider range of LLM backbones and multimodal architectures, and ensuring fair comparisons within identical settings. Furthermore, comparisons with more recent state-of-the-art algorithms are necessary to demonstrate effectiveness. Reviewer opN4 questioned whether the detailed design represents an optimal solution, suggesting it may limit deeper cross-modal alignment. The manuscript requires careful polishing to clarify the motivation and make the narrative more convincing. Given these substantial concerns, the paper cannot be accepted at this time. We encourage the authors to address these comments fully before submitting the work elsewhere.

**Reviewer Concerns:**

Through the rebuttal, the authors have successfully addressed several concerns regarding design motivation, paper presentation, and specific experimental comparisons. However, the experimental evaluation remains a critical shortcoming. Despite the revisions, the additional comparisons and detailed evaluations provided are unconvincing and insufficient, necessitating further investigation and validation.

**Reviewer Scores:**

The reviewers would be satisfied with the author's commitment to paper presentation and several comparisons and discussions in the revised version, while they may still hold concerns regarding the provided experimental evaluations and in-depth analysis.

---

### Decision · Program_Chairs · 2026-01-26

Reject